# Effect of Er:YAG Laser Exposure on the Amorphous Smear Layer in the Marginal Zone of the Osteotomy Site for Placement of Dental Screw Implants: A Histomorphological Study

**DOI:** 10.3390/jfb14070376

**Published:** 2023-07-18

**Authors:** Nikolay Kanazirski, Diyana Vladova, Deyan Neychev, Ralitsa Raycheva, Petya Kanazirska

**Affiliations:** 1Department of Oral Surgery, Faculty of Dental Medicine, Medical University-Plovdiv, 4000 Plovdiv, Bulgaria; deyan.neychev@mu-plovdiv.bg; 2Department of Veterinary Anatomy, Histology and Embryology, Faculty of Veterinary Medicine, Trakia University, 6000 Stara Zagora, Bulgaria; diyana.vladova@trakia-uni.bg; 3Department of Social Medicine and Public Health, Faculty of Public Health, Medical University-Plovdiv, 4000 Plovdiv, Bulgaria; dirdriem@gmail.com; 4Department of Imaging Diagnostics, Dental Allergology and Physiotherapy, Faculty of Dental Medicine, Medical University-Plovdiv, 4000 Plovdiv, Bulgaria; p.hadzhigeorgieva@mu-plovdiv.bg

**Keywords:** Er:YAG laser, implantology, amorphous smear layer, histomorphological study

## Abstract

The placement of dental screw implants typically involves the use of rotary techniques and drills to create a bone bed. This study explores the potential benefits of combining this method with an Er:YAG laser. Split osteotomies were performed on 10 jaws of euthanized domestic pigs (Sus scrofa domestica), with 12 mandibular implant osteotomies in each jaw, divided into 4 groups. In order to make a comprehensive assessment of the effect of Er:YAG lasers, histomorphological techniques were used to measure the reduction in amorphous layer thickness after Er:YAG laser treatment, both with and without the placement of dental screw implants from different manufacturers. Following bone decalcification and staining, the thickness of the amorphous layer was measured in four groups: Group A—osteotomy performed without Er:YAG laser treatment—had amorphous layer thicknesses ranging from 21.813 to 222.13 µm; Group B—osteotomy performed with Er:YAG laser treatment—had amorphous layer thicknesses ranging from 6.08 to 64.64 µm; Group C—an implant placed in the bone without laser treatment—had amorphous layer thicknesses of 5.90 to 54.52 µm; and Group D—an implant placed after bone treatment with Er:YAG laser—had amorphous layer thicknesses of 1.29 to 7.98 µm. The examination and photomicrodocumentation was performed using a LEICA DM1000 LED microscope (Germany) and LAS V 4.8 software (Leica Application Suite V4, Leica Microsystems, Germany). When comparing group A to group B and group C to D, statistically significant differences were indicated (*p*-value = 0.000, *p* < 0.05). The study demonstrates the synergistic effects and the possibility of integrating lasers into the conventional implantation protocol. By applying our own method of biomodification, the smear layer formed during rotary osteotomy can be reduced using Er:YAG lasers. This reduction leads to a narrower peri-implant space and improved bone-to-implant contact, facilitating accelerated osseointegration.

## 1. Introduction

Restoring the function and esthetics of the dentition has become an attainable goal in modern dentistry thanks to the major advancements in dental implantology. Placing dental implants has become a routine procedure in modern dental practice. A thorough understanding of the mechanisms of osseointegration and the factors influencing this process is essential to achieve positive outcomes in implantological treatment. The term osseointegration was introduced by Brånemark in 1969. Osseointegration was initially defined at the light microscopic level as “a direct structural and functional connection between living bone and surface living bone and the surface of a loadbearing implant” [1]. A large number of authors have worked on the mechanisms of osseointegration [2,3]. In recent years, the definition of osseointegration has changed, with the following definition being formulated: “a process in which a clinically asymptomatic rigid fixation of alloplastic material is achieved and maintained in bone during functional loading” (Brånemark 1985) [1,4].

Osseointegration is the direct structural and functional connection between living bone tissue and the implant, and it is considered to be an indicator of the success of implant placement. Osseointegration represents a specific stage of the evolution of the implant–tissue interface that results from the interaction of bone and bioinert materials [5].

To describe the complex interaction of biomaterials and tissues, Kasemo and Lausmaa used the term “implant-tissue interface”, which refers to a qualitatively new structure formed as a result of the interaction between the materials and the biological system [6]. 

The process of osseointegration depends on multiple factors that act during the four phases of the healing process. The framework used for the phases of osseointegration was defined based on an interpolation of the concept expressed by Stadelmann et al. [7] regarding the physiology and healing of chronic bone wounds. There are four phases of the healing process after implant placement: hemostasis, inflammation, proliferation and remodeling. According to a study by Terheyden et al. [8], the first signs of remodeling after dental implant placement were observed after about 6 weeks, which was two weeks later than in an animal model. The formation of titanium oxide on the implant surface after placement creates a prerequisite for osseointegration [4]. Ti-alloys are popular candidates for load-bearing implant applications owing to their high strength-to-weight ratio, good biocompatibility and excellent corrosion resistance, regardless of whether they are a pure Ti, Ti-6Al-4V, Ti-Nb alloy or any other alloy. Chakkravarthy et al. reported that Ti-30Nb-2Zr, which is a promising next-generation biomedical implant material, was fabricated in the form of a porous scaffold using the selective laser melting (SLM) additive manufacturing route [9,10]. Titanium nanoparticles (Ti-NP), as a material with suitable mechanical properties, can be a used for the production of dental ceramics with higher strength properties via combination with lithium disilicate [11].

According to Brånemark [1], osseointegration refers to a phenomenon in which the implant becomes so ankylosed with the bone that they cannot be separated without fracture.

Various factors that influence the rate and success of osseointegration can be categorized as those related to implant characteristics, such as the physical and chemical macro- and micro-design of the implants; as bone characteristics, such as the quantity and quality of bone and the local and systemic conditions of the patient; or the time and protocol used for the functional loading of the dental implant [12]. To ensure proper healing of the placed implant, it is necessary to guarantee good primary stability. Another important aspect is the need to control micro-movements. Primary stability is defined as the biometric stability immediately after implant placement, and it is a direct result of mechanical engagement of the implant with the adjacent bone [13]. It is a result of the frictional interaction between the implant and the bone. The formula defining this interaction is as follows:
F = k·N, as:
F—the force of friction; k—coefficient of friction (surface-specific); and N—pressure between the two surfaces (Joos) [14]. Factors that increase k and N lead to an increase in primary stability [15].

Implant surface design has evolved to meet oral rehabilitation challenges in both healthy and compromised bone. Many studies aim to comprehensively discuss currently available implant surface modifications that are commonly used in implantology in terms of their impact on osseointegration and biofilm formation, which is critical in helping clinicians to choose the most suitable materials to improve the success and survival of implantation [16].

The use of implants with modified surfaces aims to accelerate osseointegration, which is evidenced by achieving control of the stability of the placed implants. The microrelief of the implant surface determines the good connection of the implant surface to the fibrin skeleton, as well as the course of contact osteogenesis and the increase in the total area of the region of the implant–tissue interface. Modification methods are divided into two main groups: subtractive methods, in which defects on the implant surface are created by removing an amount of material, and additive methods, in which positive roughness is created by adding material.

Subtractive methods are as follows: machining, etching, abrasive jet methods, SLA (Sandblasted with Large grift Al_2_ O_3_, followed by Acid etching) and laser texturing.

Additive methods are as follows: titanium plasma spray, hydroxyapatite coating, anodic oxidation and brushite coating.

Changing the surface chemistry of implants involves a classic combination of abrasive jet treatment and subsequent high temperature etching.

Modern dentistry faces the challenge of providing rapid treatment results within a short period without compromising quality. The placement of dental implants typically involves the use of rotary techniques and drills to create a bone bed for implant placement. During this procedure, water cooling is used, and the revolutions per minute (RPM) value is carefully chosen to avoid overheating the bone and denaturing its protein structures. In addition, the use of a rotary technique to create a bone bed results in a smear layer, which can affect primary stability and may lead to changes even beyond day 12. Following the introduction of Er:Yag lasers, the rapid development of surgical technologies has enabled the creation of implant sites using alternative techniques, such as laser surgery. However, attempts to use lasers alone for osteotomy have revealed certain drawbacks, including prolonged treatment duration and inaccuracies in calibrating the osteotomy hole. Nevertheless, using the Er:Yag laser at a wavelength of 2940 nm offers benefits such as proper decontamination of tissues, the absence of a smear layer on the osteotomy surface and reduced bleeding. These advantages provide opportunities for accelerated osseointegration and early prosthetic treatment.

The combination of conventional rotary techniques and laser osteotomy demonstrates synergistic effects, making it possible to integrate lasers into the conventional implantation protocol. Early loading of dental implants placed using the two-stage technique becomes feasible through the combined use of rotary and laser techniques. This combined technique aims to create the conditions required for accelerated osseointegration, which is crucial to improving masticatory function and facilitating esthetic rehabilitation for the patient.

When placing a screw dental implant, proper preparation of the recipient site is crucial. The method of recipient site preparation determines the possibility of early loading based on the achieved primary stability of the implants. This stability depends on both the type of bone located at the implantation site (according to the Misch classification) and the technique used. Several approaches can be employed to attain primary stability. The conventional and most commonly used method is the rotary technique with water cooling.

In several in vitro studies using bovine models, it has been reported that one of the major factors influencing successful osseointegration is the temperature generated during recipient site preparation. The authors discuss the importance of not exceeding the temperature limit of 47 °C to prevent bone necrosis [1,17]. Various measures are being explored to reduce heat generation during osteotomy, including drill geometry and design, bone density and cortical thickness, single-stage or reciprocating drilling, use of reusable drills, internal or external cooling and the pressure applied by the operator during osteotomy [18].

The use of ultrasound for osteotomy has the disadvantage of being relatively slower than the standard rotary technique. However, one benefit of using ultrasound is that it enables much faster tissue regeneration than the conventional technique [19].

Laser technology provides another option for preparing the implant site. Among all types of lasers, the Er:Yag laser is the most suitable for this purpose. This type of laser allows us to work on bone structures without causing carbonization due to its pulse operation and effective cooling during the procedure [20]. The selection of the correct wavelength and laser parameters is of paramount importance in terms of minimizing the risk of thermal damage to the bone [20,21]. According to studies conducted by some authors, the three aforementioned techniques used to preparing the implant site are comparable, at least during the early stage of the healing process [22,23]. Achieving temperature control via various osteotomy techniques can be challenging. All of these factors contribute to the possibility of influencing the duration of osseointegration [24].

### 1.1. Changes in Alveolar Bone When Using Different Osteotomy Techniques

Understanding the biological properties of alveolar bone is crucial to achieving successful outcomes during surgical procedures that involve dental implant placement or the use of bone substitutes. Bone is a specialized connective tissue characterized by a highly mineralized extracellular matrix. It consists of cells and an extracellular matrix comprising collagen fibers and non-collagen proteins, with its hardness attributed to the inorganic component that fills the protein matrix. Anatomical variations in the volume of the substantia compacta also influence the feasibility of implantation. All of these factors play a significant role in the success of osseointegration, depending on the osteotomy technique employed.

A fundamental principle in dental implantology is that a less traumatic surgical procedure and causing minimal injury (surgical trauma) to the bone tissue at the recipient site during implantation facilitate faster new bone formation on the implant surface [25].

### 1.2. Rotary Technique

The conventional approach used in dental implant placement involves the use of rotary instruments to prepare the implant site. Several factors must be considered when employing this type of instrument. Firstly, the number of implants placed using these drills should be monitored, as worn-out drills can lead to the overheating of bone. Secondly, the frictional movements of the instruments must be controlled to ensure optimal cutting of the bone tissue while avoiding excessive pressure on the tip.

After drilling with a rotary instrument, the bone surface exhibits a uniform structure covered with an amorphous smear layer of varying thickness. This layer can act as a barrier, impeding the interaction of blood components with the underlying tissue and potentially delaying the healing process [26].

Failure to meet the aforementioned conditions can result in bone overheating, leading to protein denaturation, enzyme inactivation, osteoblastic and osteoclastic necrosis and bone resorption. These factors can disrupt bone–implant integration and eventually lead to implant osseointegration failure [18,27]. According to several authors, controlling the heat generated during the procedure contributes to predictable and successful osseointegration [18]. Studies conducted by Barrak et al. indicate that drill wear varies based on the rotation speed used. Noticeable wear occurred after 210 osteotomies at 800 rpm, 120 osteotomies at 1200 rpm and 90 osteotomies at 1500 rpm [28]. Other authors have focused on drill design, which they believe plays a significant role in heat generation during osteotomy [24,29,30]. The impact of the pressure exerted by the operator on the heat-generating tip has been underestimated in many studies. According to Eriksson and Adell [31], light pressure is necessary, though defining the precise numerical value for the applied force is challenging to determine. Typically, light hand pressure falls within the range of approximately 2 kg/cm^2^. Further research is required to elucidate the significance of operator-applied pressure on the handpiece and heat generation [32].

### 1.3. Laser Technology

Laser technology is distinct and versatile due to its physical characteristics. Properties such as collimation, coherence and monochromaticity are achieved through high-energy processes within the optical cavity. The laser beam emitted from this cavity possesses these properties when it interacts with the patient’s tissues. Although the fundamentals of lasers may not be intuitively clear to most clinicians, their applications in medicine have expanded since Mainman developed the first ruby laser in 1960.

In recent years, scientific advancements have introduced an additional tool that is increasingly being incorporated into the clinical practice of dental implantology—the Er:YAG laser. This precise instrument is used for bone ablation, as its near-infrared wavelength of 2940 nm is highly absorbed by water and, to a much lesser extent, by hydroxyapatite [33,34]. The laser is a suitable instrument for effectively ablating bone through microexplosions, with minimal carbonization of the surrounding and underlying tissues occurring [35]. Bone treatment using an Er:YAG laser is performed with a simple non-contact metal tip and water irrigation. The most commonly used parameters are a 2-millmeter spot size, 500–1000 mJ per pulse, a 400-millisecond pulse duration and fluence of 16–32 J/cm^2^. External cooling with saline or sterile distilled water during the procedure minimizes bone carbonization and improves the healing process. Some authors have concluded that the Er:YAG laser can stimulate new bone growth around titanium implants and promote better osseointegration than conventional osteotomies [36].

Other studies have reported similar laser-induced stimulation of bone growth with minimal changes and limited damage, albeit restricted to a superficial amorphous layer of bone approximately 30 µm in width. The absence of a broad smear layer in laser-treated bone could potentially enhance the adhesion of blood components in the early stages of the healing process [26]. The smear layer can act as a barrier, impeding the interaction between blood components and the underlying bone, thus delaying the healing process [37].

Through scanning electron microscopy (SEM) examination of bone surfaces treated with the Er:YAG laser, a fibrin-like tissue covering the cavity surface was found. This material cannot be removed via washing with saline and likely acts as a mechanical valve for plasma proteins, providing a base for adherence to the fibrin clot. In vitro experiments on rat calvaria by Sasaki et al. suggested that this phenomenon is the reason for the accelerated healing process. The authors did not observe this phenomenon in bone tissues treated with rotary instruments [26]. O’Donnell et al. reported that the faster rate of bone formation in laser-induced bone growth stimulation may allow earlier functionality and loading of the implant [38].

Sasaki et al. performed Fourier-transform infrared (FTIR) spectroscopy on bone samples. Raw bone exhibited bands of orthophosphate, carbonate and hydroxyl groups. The spectra of the water-cooled Er:YAG laser-ablated samples showed similar bands, albeit with decreased amides and hydroxyl groups at the expense of increased orthophosphate. FTIR observation demonstrated that Er:YAG laser-treated surfaces were composed of more inorganic structures than organic structures, in contrast to standard-treated bone surfaces. Spectroscopic analysis did not detect any traces of toxic substances or their effects, if any, on the bone surface [26].

Accurate selection of the appropriate laser system and wavelength, along with proper cooling, is essential to control tissue and implant temperature changes during laser treatment [39,40].

The advantages of using the Er:YAG laser include the absence of vibrations and the provision of detoxification and bactericidal effects. Additionally, bleeding and the possibility of injury to adjacent tissues are reduced. This instrument offers accuracy and precision in cutting. Other benefits of using this type of laser include reduced trauma generation during the procedure and fewer post-operative complications [21,41,42]. However, the routine use of lasers for bone ablation is currently limited by technical shortcomings, such as a lack of depth control and difficulties in safely targeting the laser beam [43]. Consequently, inaccuracies in the calibration of the implant site may arise. Stubinger et al. provide evidence of the slightly angular and irregular shape of the cavity with inaccuracies in its height and diameter [22]. To overcome these shortcomings, Seymen et al. utilized stereolithographic surgical guides after conducting a 3D analysis to shape the implant site. They then performed a new 3D analysis of the shaped site to compare the data. Although the results obtained were encouraging, statistically significant differences in diameter and length were found [44].

Several studies have demonstrated that the Er:YAG laser can cut bone precisely, with minimal thermal damage of only 10–15 μm [45]. This result is due to its higher water absorption coefficient, which is up to 15,000–20,000 times that of other lasers. The laser removes a fixed amount of material per pulse, allowing precise control of the cutting depth [46]. Furthermore, laser bone ablation is a non-contact procedure, eliminating the need to apply pressure during operation and making lasers superior to mechanical drilling [47,48].

Regarding the biological effects of Er:YAG lasers, there are some controversies in the literature. According to an early study by Lewandrowski, Er:YAG laser ablation demonstrated a loss of organic matrix and biological activity that negatively affected guided tissue regeneration. However, the authors suggested that the free exposure of bone minerals in the modified surface layer after laser treatment might be essential to stimulating bone regeneration and faster osseointegration [49].

Kesler et al. also stated that Er:YAG lasers can be clinically used to prepare the implant site, yielding better results in terms of osseointegration and bone healing than conventional methods. In their histological assessments, the authors found that laser-prepared bone had a higher percentage of BIC (Bone Implant Contact) at the implant interface than rotary-prepared bone at week 3 (Er:YAG: 59.48%; drill: 12.85%) and week 12 (Er:YAG: 73.54%; drill: 32.65%) [50]. Similar results have been reported by other authors [51]. In recent years, studies in dental implantology using the Er:YAG laser have demonstrated an extremely favorable effect on osseointegration. Rapid healing processes and accelerated osseointegration have been observed [39,52].

Er:YAG laser osteotomy is a non-contact procedure, which is free from mechanical vibrations and bone fragments, and it provides an aseptic surgical field. Consequently, rapid healing is achieved without swelling, inflammatory complications and severe pain. Intraoperative surgical complications, such as bone fracture, nerve involvement, dislocations or damage to adjacent teeth, are extremely rare when lasers are used. The Er:YAG laser represents a minimally invasive method that helps to reduce late complications. This new dentistry technology utilizes the energy of light to eliminate bacteria during treatment, thereby ensuring fewer or no post-operative complications and infections. A significant advantage of Er:YAG lasers is their ability to differentially intervene on bone or soft tissues. This step is carried out by changing the mode from the control panel.

The combination of the conventional rotary technique and laser osteotomy demonstrates synergistic effects. Osteotomies performed on domestic pig mandibles show a significant reduction in the amorphous layer on the cavity surface treated with an Er:YAG laser at a wavelength of 2940 nm when using an original method. The optical characteristics of the wavelength, along with the absorption of the laser beam by water molecules instead of the bone’s hydroxylapatite, result in no thermal damage. Proper cooling with serum further contributes to the absence of thermal damage [29,53].

Some authors have concluded [27,49] that the laser can stimulate new bone growth around titanium implants and promote better osseointegration than conventional techniques.

## 2. Objective

The objective of this study was to compare the smear layer thickness of an osteotomy produced using a conventional rotary technique and an osteotomy additionally treated using an Er:YAG laser, according to the method that we previouslydeveloped.

## 3. Materials and Methods

### 3.1. Materials

Mandibles from 10 domestic pigs (Sus scrofa domestica) were examined. The bone biospecimens were obtained from a regulated slaughterhouse immediately after the animals were euthanized. Treatment was conducted immediately to ensure that the bone still had the characteristics of viable bone. The lower edge of the mandible was used, as it corresponded to the alveolar ridge in an edentulous jaw. The soft tissues and periosteum were first removed from the lower edge of the jaws.

### 3.2. Study Design

#### 3.2.1. Mandibular Osteotomy

Biospecimens were divided into four experimental groups. After removing the soft tissue and periosteum, 30 osteotomies were performed in each of the 4 experimental groups using standard osteotomy drills for 3.75 mm × 8 mm spiral implants of the Alpha-Bio Neo system (Petach Tikva, Israel) and 3.25 mm × 8.5 mm implants of the BT Konic (BTK) system (Vicenza, Italy).

These drills were designed to be active yet gentle on the bone, having an active apical tip and traction wings, providing optimal primary stability while maximizing bone volume preservation. They were made of pure grade four titanium. This process provided high technological characteristics of strength and durability. The surface was thermally etched after sandblasting to create optimal porosity, and it is called Nano Tec.

This approach provides the following benefits:-Improved early bone–implant contact, which is an important factor for excellent primary stability;-Long-term bone–implant contact;-Accelerated and improved osseointegration;-Increased secondary stability.

The implants used were recommended for all types of bone.

Standard preparation of the osteotomy using the implantology surgical set of the respective system was carried out. The diameter of the final osteotomy drill was 0.1–1.2 mm smaller than the diameter of the implant used. Trepanations were performed using a Bien Air Chiropro (Bien-Air Dental SA, Bienne, Switzerland) implantology unit, which is a reduction implantology handpiece that undergoes continuous external cooling with 0.9% sterile NaCl solution.

The sequence of bone cavity preparation was as follows:-Marking the location to place the implant on the cortical bone using a round bone cutter.-Trepanning the bone with a pilot cutter to pre-determine the length of the implant, followed by a depth gauge check.-Preparing the implant site using cutters with successively increasing diameters to a diameter of 0.1–1.2 mm smaller than that of the implant. The speed of rotation of the tools was 600–800 revolutions per minute.-Taking in the cortical bone phase with the corresponding profile cutter.

After the cavity was finally prepared, the surface treatment of the walls with the surface treatment module began, which started with the use of an Er:YAG laser at a preferred wavelength of 2940 nm. The Er:YAG laser (also called an erbium-doped yttrium aluminum garnet laser or erbium YAG laser) is a solid-state laser with erbium-doped yttrium aluminum garnet (Er:Y3Al5O12). Er:YAG lasers typically emit infrared light. For the purposes of this study, treatment was performed using a “non-contact granulation tissue ablation” program.

Implant site preparation using the standard rotary technique was performed within the normal time frame. Surface treatment with the Er:YAG laser took two-to-three minutes per osteotomy opening, which was not a statistically significant prolongation compared to laser preparation alone, which took 25 min on average.

In group A, no additional treatment of the trepanation hole was performed.

In group B, the osteotomy surface was treated with an Er:YAG laser LiteTouch (Light Instruments Ltd., Yokneam Illit, Israel) at a wavelength of 2940 nm. The Granulation Tissue Ablation Non-Contact program was used for the treatment, using the following parameters:Laser energy: 400 mJ;Pulse frequency: 17 Hz;Water spray level: 6;Power: 6.80 W.

An AS 7631 (X) Side Firing Tip with a diameter of 1.3 mm and a length of 19 mm was used. It radiated energy at 90° to the longitudinal axis and within a 180° perimeter. Therefore, the treatment was carried out in two quadrants—the medial and distal quadrants (vestibular and oral). Treatment started from the inside and evenly continued toward the surface, with light rotary movements, first in one quadrant, then in the other quadrant, for about 2–3 min. In this way, the entire cavity surface was treated from the bottom to the surface (Figure 1a,b).

In group C, after osteotomy was performed using a standard rotary technique, titanium alloy implants with an etched surface, which had diameters of 3.75 mm and 3.25 mm and lengths of 8 mm and 8.5 mm, respectively, which were produced by the above-mentioned companies, were manually placed.

In group D, implants of the same types and sizes were placed after treating the cavity walls with an Er:YAG laser as in group B.

#### 3.2.2. Histological Techniques

The mandible specimens were cut around the osteotomies to obtain 1 × 1 cm cubes, which then underwent the following treatment stages (a routine methodology described by Dyakov et al., 1989) [54]:*Decalcification*

During decalcification, calcium was removed from the bones. The process involved the following steps:

Fixing the material in 10% formalin for two-to-three days, before rinsing it in running water and placing it in a decalcifying liquid, which was obtained by adding 100 cm^3^ of distilled water to 5–7 cm^3^ of concentrated nitric acid. The liquid was shaken several times every day to release the carbon dioxide bubbles that formed. The duration of decalcification depended on the size and density of the object. The material was considered decalcified when it could be easily cut with a knife without resistance or crunching sounds. The decalcified material was transferred to a 5% solution of sodium sulfate for twenty-four hours and rinsing in running water for twenty-four-to-forty-eight hours.
*Embedding the material in paraffin blocks*

Next, we dehydrated the fixed material in ascending series of alcohols—80%, 90% and 95% absolute ethyl alcohol. The material stayed in each alcohol for twenty-four hours; after that step, we successively transferred the material to three glass jars containing pure benzene for 15 min. The next step involved transferring the material to a mixture of paraffin and benzene (in equal parts) for an hour and a half at a temperature of 37 °C in a thermostat for thirty minutes and washing the blocks (impregnation) in pure molten paraffin in a thermostat at 58–60 °C for five-to-six hours to completely remove the benzene.

The tissue was immersed in clean, melted paraffin at 56 °C and left to harden at room temperature. Paraffin blocks were then attached to wooden blocks using a drop of melted paraffin and cut using a microtome. Sections obtained using the paraffin microtome were picked up using a brush and placed in warm water (37–40 °C) to stretch. Afterward, they were mounted on coverslips pre-coated with a thin layer of a mixture of equal parts of glycerin and albumin. The coverslip with the section was placed onto a slide, covered with filter paper and gently pressed to remove excess liquid. The slide was left to dry and adhere for twenty-four hours.
*Hematoxylin and eosin staining*

The staining process was carried out as follows:

We deparaffinized the sections sequentially in xylene and benzene for three-five minutes, before hydrating them in a series of alcohols with decreasing concentrations—absolute alcohol 96%, 90%, 80%, 70% and 60%. We then stained the sections in hematoxylin for two-to-ten minutes. We rinsed the sections in water and placed them in tap water for fifteen-to-thirty minutes until the sections acquired a light violet-to-light blue color. The sections were stained with eosin for thirty-to-forty seconds, and we then rinsed them in water and dried them on filter paper.

We dehydrated the sections in a series of alcohols with increasing concentrations—60%, 70%, 80%, 90%, 96% and absolute alcohol for one minute in each jar. We then lightened the sections in xylene for two-to-three minutes and mounted them in Canada Balsam.

A large number of permanent slides with horizontal and vertical sections were prepared, and from these samples, slides used for histological analysis were selected. In all groups, measurements of the amorphous layer on the surface of the trepanation hole were made in vertical sections at three positions—apical, median and marginal positions. The examination and photomicrodocumentation were performed using a LEICA DM1000 LED microscope (Germany) and LAS V 4.8 software (Leica Application Suite V4, Leica Microsystems, Germany). A metric measurement of the smear layer was made, which was based on the differences between hematoxylin and eosin staining. Healthy bone and the smear layer area have different staining intensities and clear boundaries. Statistical analysis were performed based on these measurements.

Histological preparations and analysis were conducted at the Faculty of Veterinary Medicine, the Department of Veterinary Anatomy, Histology and Embryology and the Department of General and Clinical Pathology, Trakia University.

### 3.3. Statistical Analysis

Standard descriptive statistics were used to present the quantitative variables in terms of their mean and standard deviation (mean ± SD), and the Shapiro–Wilk test was applied to inform the distribution of the units of observation included in the sample. Comparisons between two groups were analyzed via Student’s *t*-tests for independent samples, and comparisons between more than two groups were analyzed via one-way ANOVA with Bonferroni correction for multiple comparisons. A 2-sided *p*-value of <0.05 was considered statistically significant. Statistical analyses were performed using SPSS Statistics v. 26 software (IBM Corp. Released 2019. Armonk, NY, USA).

## 4. Results

### 4.1. Histological Analysis

Group A—The survey of horizontal and vertical histosections located across the contact surface of the bed showed a rough bone surface with an irregular periphery along the incision edges and numerous microcracks resulting from the osteotomy. The cracks were filled with bone fragments and soft tissue, collectively forming an amorphous layer that covered the trepanation surface. The amorphous layer blocked the Volkmann’s and Haversian canals. The osteotomy edges beneath the amorphous layer displayed varying degrees of destructive changes, resulting in a porous surface with low-grade-to-absent thermoalteration (Figure 2).

Group B—The survey of histosections across the contact surface of the implant site revealed linear and distinct trepanation edges, which were free of bone and soft-tissue fragments. Volkmann’s and Haversian canals were open. The amorphous layer was irregular, vague or fragmented, while, in some places, it was completely absent. This layer can be further subdivided into two distinct sublayers—the superficial layer (PL) and the deep layer (DL). The superficial layer showed signs of carbonization in certain areas, along with remnants of bone and soft tissue. The deep layer exhibited mild traumatic damage that resulted from the preliminary mechanical treatment (Figure 3).

Group C—The histological findings were almost identical to those observed in group A. The main difference was the smaller thickness of the amorphous layer on the contact surface due to the slight compression that occurred during implant placement. As an exception, there were certain areas on the surface that exhibited characteristics of compression trauma, which were included in the amorphous layer. The boundaries of this layer were not clearly defined (Figure 4).

Group D—The histological findings were relatively similar to those of experimental group B. Smooth linear edges were predominantly observed adjacent to the trepanation surface. No isolated and/or layered amorphous masses of bone or soft-tissue fragments were found. The surface layer appeared to be wel structured, with almost no signs of compression trauma, alteration and carbonization. The Volkmann’s and Haversian canals extended directly to the surface, allowing direct contact between the cells within them (including osteoblasts and osteoclasts) and the implant surface. The thickness of the amorphous layer was extremely small, and in many areas, it was practically absent (Figure 5).

Implant site preparation using the standard rotary technique was performed within the normal time frame. Surface treatment with the Er:YAG laser took two-to-three minutes per osteotomy opening, which is not a statistically significant prolongation compared to laser preparation alone, which took 25 min on average. No signs of bone carbonization, melting or cracking were observed in any of the groups.

Histomorphological measurements showed the following amorphous layer thicknesses by group: group A—21.813 to 222.13 µm; group B—6.08 µm to 43.64 µm; group C—5.90 to 54.52 µm; and group D—1.29 to 7.98 µm.

### 4.2. Statistical Analysis

Statistical comparison between groups A and B and groups C and D was performed. Comparisons between two groups were analyzed VIA Student’s *t*-tests for independent samples. Surface deformations of the implant site were conventionally divided into three locations: the apical, median and marginal parts. In all three parts, significant differences in deformations were observed among the compared groups (*p*-value = 0.000, *p* < 0.05).

Descriptive and inferential statistics of deformation in implant cavity—standard drills (in microns) and deformation in implant cavity—using the Er-YAG laser (in microns) by location—apical, median and marginal—are presented in Table 1. 

For all locations, using the Er-YAG laser resulted in significantly lower deformations in the implant cavity—approximately three times lower average values were recorded compared to the standard drill means (Table 1). One-way ANOVA analysis demonstrated statistically significant differences in the deformation in implant cavity between the apical and marginal locations when the standard drill technique was applied (Bonferroni post-hoc test, *p* = 0.004). The Er-YAG laser mean values of deformation by location were consistent, and no statistically significant differences were observed (Figure 6).

Descriptive and inferential statistics of histomorphological measurements in group A and group B are presented in Table 2.

For all locations, mean histomorphological measurements in group B were significantly lower than the average scores obtained in group A (Table 2). One-way ANOVA analysis indicated statistically significant differenced in mean scores of histomorphological measurements—in group A, these differences were between the apical and marginal locations (Bonferroni post-hoc test, *p* = 0.003), and in group B, these differences were between the average scores in marginal and both apical and median locations (Bonferroni post hoc test, *p* = 0.000 and *p* = 0.001, respectively) (Figure 7). The median values of the histomorphological measurements for all locations observed in group B were fairly homogeneous, in contrast to the median values for the same locations measured in group A (Figure 7).

Descriptive and inferential statistics of histomorphological measurements in group C and group D are presented in Table 3.

For all locations, mean histomorphological measurements in group D were significantly lower than the average scores obtained in group C (Table 3). One-way ANOVA analysis indicated statistically significant differences in the mean scores of histomorphological measurements in both group C and group D between apical and marginal locations (Bonferroni post hoc test, *p* = 0.003 and *p* = 0.031, respectively), as shown in Figure 8. The median values of the histomorphological measurements for all locations observed in group D were fairly homogeneous, in contrast to the median values for the same locations measured in group C (Figure 8).

## 5. Discussion

This study was designed to provide evidence of the role of Er:YAG laser treatment in implant site preparation. The combination of conventional rotary osteotomy with additional laser cleaning of the smear layer demonstrated a synergistic action. Rotary instruments have no detrimental effects on the viability and differentiation of cells that form new tissue and ensure implant osseointegration. However, the groups treated with the Er:YAG laser at a wavelength of 2940 nm demonstrated the possibility that bone ablation and cleaning of the amorphous layer could occur without thermal side effects affecting the surrounding tissues. Sasaki et al. [26] performed a similar analysis of the ultrastructure of the parietal bone of Wistar rats treated using an Er:YAG laser, a CO_2_ laser and a conventional drill. Microscopy showed that the Er:YAG laser osteotomy resulted in the layer changing from 13.2 to 30 µm. The surface layer had numerous microcracks. The deep layer was less affected, having few microcracks, and lacked the production of toxic substances often observed after irradiation via other lasers [39]. The results of our study correspond to these results. For all locations, using an Er-YAG laser resulted in significantly lower deformations in the implant cavity—approximately three times lower average values than those of the standard drill means were recorded. For all locations, mean histomorphological measurements in group B were significantly lower than the average scores obtained in group A. For all locations, mean histomorphological measurements in group D were significantly lower than the average scores obtained in group C.

Lewandrowski et al. [49] reported that bone healing was faster in Er:YAG laser osteotomies than in standard procedures due to the absence of a smear layer. Similar results were reported by Kesler et al. [50] after Er:YAG laser preparation of implantation sites in rat tibiae. This preparation process resulted in even better direct bone-to-implant contact (BIC). el-Montaser et al. confirmed faster osseointegration and new bone formation in implant canals prepared using the Er:YAG laser than in those prepared using standard techniques. The Er:YAG laser did not compromise bone healing and the subsequent osseointegration of the dental screw implants; on the contrary, it stimulated both factors [55].

Schwarz et al. investigated the width of the peri-implant gap and the bone-to-implant contact in osteotomy sites prepared using the Er:YAG laser, CO_2_ laser and standard instruments in four beagle dogs. The authors performed a histomorphometrical assessment of the osseointegration of titanium dental implants in the three groups. Despite the wider peri-implant gaps identified during the placement in the Er:YAG laser group, at two weeks, statistical analysis of the results showed higher values of bone-to-implant contact in this group, and at 12 weeks, complete new bone was formed around the implants [56]. Authors of other studies reported similar results [38].

We did not find existing reports regarding the combination of the standard rotary preparation technique and subsequent osteotomy surface treatment using an Er:YAG laser, which was proposed in this study, in the available literature. The proposed method optimizes the effectiveness of Er:YAG lasers in implantology.

The applied settings for biomodification of the preparation surface with the laser, along with the “tsunami” effect produced via the cooling liquid, are an original method, which aimed to achieve optimal tissue cleaning using the Er:YAG laser [57].

Based on the promising histological results obtained in our study, we plan to examine the morphological changes using scanning electron microscopy (SEM). Upon confirmation of the results, we will perform a clinical experimental study.

The synergistic combination of conventional rotary methods and Er:YAG laser treatment allows uniform implant site preparation while minimizing the risk of damage to nearby anatomical structures.

## 6. Conclusions

The smear layer acts as a barrier to the interaction of blood components in the underlying tissue with the implant surface, leading to delays and complications in the process of osseointegration. This layer mainly consists of unmineralized collagen and proteoglycans. The increase in temperature of the bone surface when using a rotary technique leads to surface carbonization and possible compromising of treatment. High hopes are placed on laser ablation and decontamination without additional heat generation due to the precise beam geometry required when using short-pulse modes of operation and copious irrigation for cooling. The reduction in the smear layer leads to tighter contact between the implant surface and the bone, resulting in better primary stability. The cleaning time is reduced, and the osseointegration process is accelerated. The Er:YAG laser program used in our study does not affect the hard tissues and, thus, does not change the dimensions of the cavity prepared using standard drills.

The potential of obtaining a smeared layer via the conventional technique and calibration inaccuracies in laser ostetomy are minimized with our proposed methodology. The study demonstrated that the combination of the rotary technique and subsequent Er:YAG laser treatment of the bone using an original method is a promising prospect in implantology, as it may allow us to achieve faster and stable osseointegration of implants, leading to early functional loading.

The synergistic effect of the combination of the two methods leads to an absolutely precisely prepared implant site, which is achieved via conventional rotary methods, and a reduced or absent smear amorphous layer on the surface via the use of laser osteotomy only.

## 7. Patent

A patent entitled “MODULAR COMPLEX FOR PREPARATION OF THE IMPLANTOLOGY BED FOR SPIRAL DENTAL IMPLANT”, which was allocated the registration № 4368 U1 by the Patent Office of the Republic of Bulgaria, resulted from the work reported in this manuscript.

## Figures and Tables

**Figure 1 jfb-14-00376-f001:**
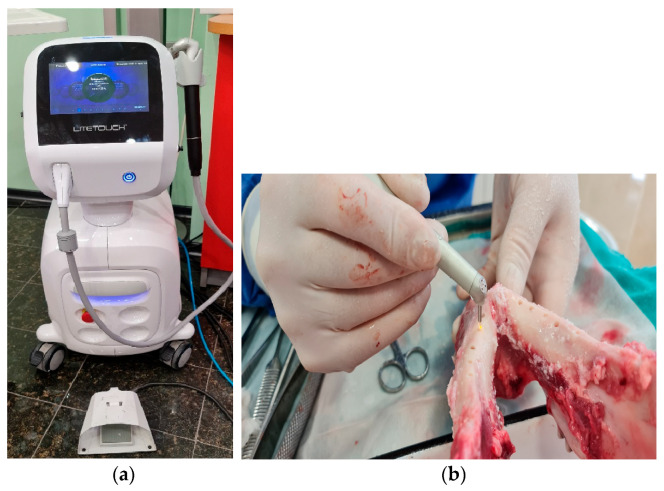
(**a**) Er:Yag laser Litetouch (Light Instruments Ltd., Yokneam Illit, Israel) with a wavelength of 2940 nm; (**b**) laser treatment of osteotomies.

**Figure 2 jfb-14-00376-f002:**
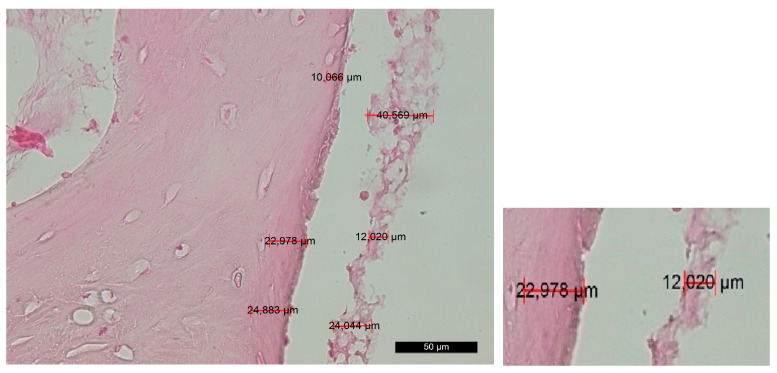
Changes in the contact surface in group A—magnification of 50 µm and an example of the measurement.

**Figure 3 jfb-14-00376-f003:**
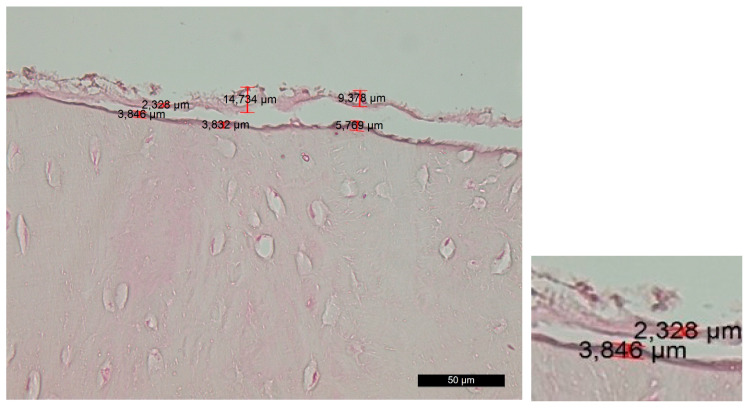
Changes in the contact surface in group B—magnification of 50 µm and an example of the measurement.

**Figure 4 jfb-14-00376-f004:**
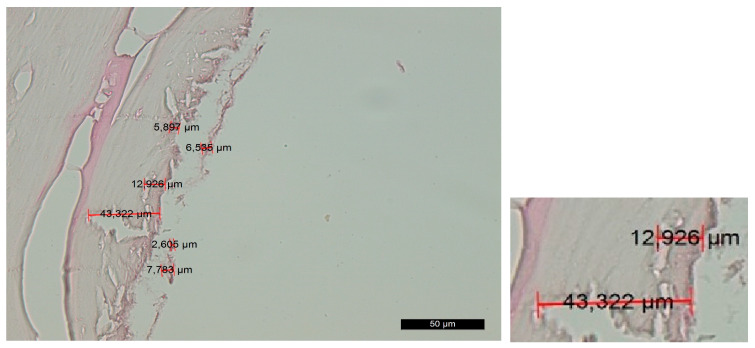
Changes in the contact surface in group C—magnification of 50 µm and an example of the measurement.

**Figure 5 jfb-14-00376-f005:**
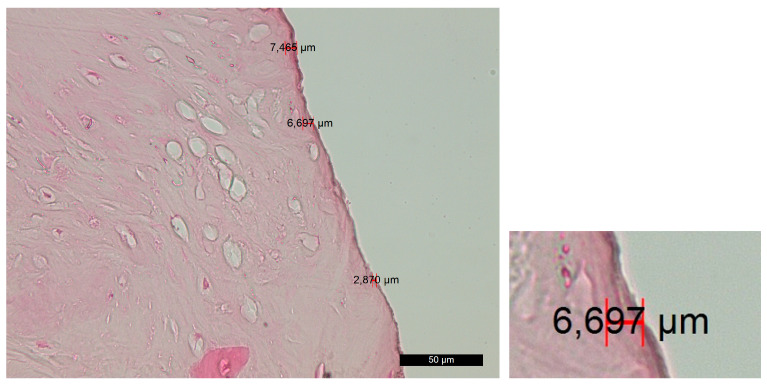
Changes in the contact surface in group D—magnification of 50 µm and an example of the measurement.

**Figure 6 jfb-14-00376-f006:**
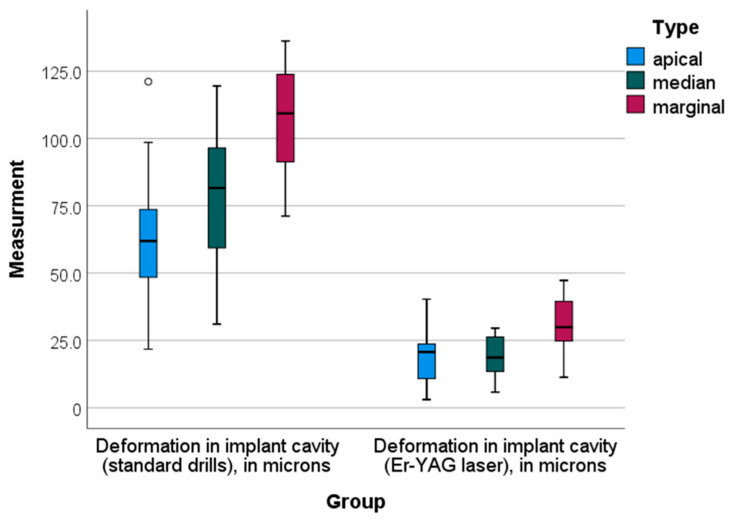
Box-plot diagram of the deformation in implant cavity by group and location.

**Figure 7 jfb-14-00376-f007:**
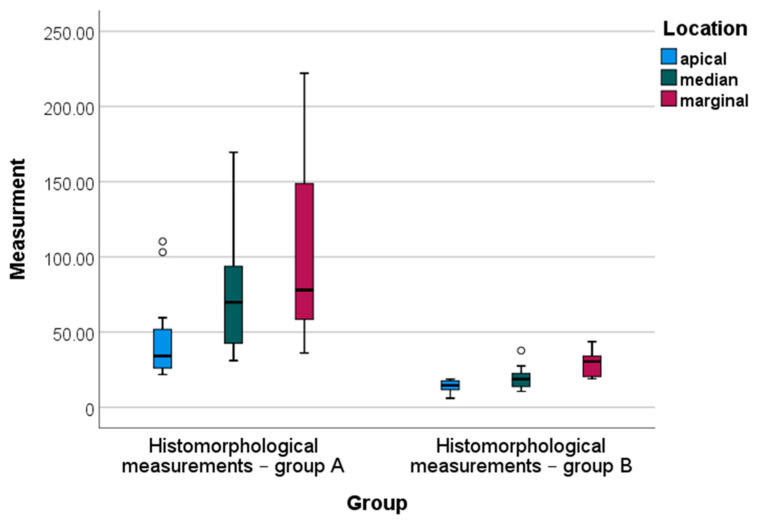
Box-plot diagram of the histomorphological measurements by group (A vs. B) and location.

**Figure 8 jfb-14-00376-f008:**
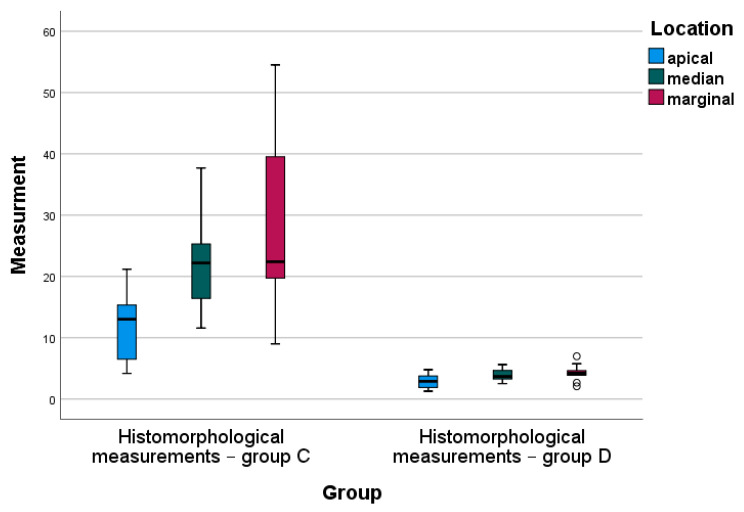
Box-plot diagram of the histomorphological measurements by group (C vs. D) and location.

**Table 1 jfb-14-00376-t001:** Descriptive and inferential statistics of the deformation measured after use of standard drill and Er-YAG laser by location.

Location	Deformation in Implant Cavity— Standard Drills (in Microns)			Deformation in Implant Cavity— Er-YAG Laser (in Microns)			*p*-Value *
	N	Mean	Std. Deviation	N	Mean	Std. Deviation	
Apical	10	64.52	28.717	10	19.50	11.142	0.000
Median	10	79.12	29.836	10	18.52	7.874	0.000
Marginal	10	106.65	20.700	10	29.52	11.533	0.000

* *t*-test.

**Table 2 jfb-14-00376-t002:** Descriptive and inferential statistics of the mean histomorphological measurements—group A vs. group B.

Location	Histomorphological Measurements— Group A			Histomorphological Measurements— Group B			*p*-Value *
	N	Mean	Std. Deviation	N	Mean	Std. Deviation	
Apical	15	44.90	27.78	15	14.24	3.73	0.000
Median	15	77.08	42.22	15	19.21	7.19	0.000
Marginal	15	104.16	61.14	15	29.00	8.08	0.000

* *t*-test.

**Table 3 jfb-14-00376-t003:** Descriptive and inferential statistics of the mean histomorphological measurements—group C vs. group D.

Location	Histomorphological Measurements— Group C			Histomorphological Measurements— Group D			*p*-Value *
	N	Mean	Std. Deviation	N	Mean	Std. Deviation	
Apical	10	12.29	5.45	10	2.84	1.11	0.000
Median	10	22.09	7.18	10	3.85	0.98	0.000
Marginal	10	27.91	13.92	10	4.29	1.40	0.000

* *t*-test.

## Data Availability

The data that support the findings of this study are available from the corresponding author upon reasonable request.

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
