# Peer review of "Effect of Er:YAG Laser Exposure on the Amorphous Smear Layer in the Marginal Zone of the Osteotomy Site for Placement of Dental Screw Implants: A Histomorphological Study"

_jfb, 2023, doi:10.3390/jfb14070376_

Round 1
Reviewer 1 Report
Authors have clearly addressed one of the common challenges in fixing dental screws over the marginal zone of osteotomy site. The paper is well written but still needs fine improvement to fit into the scope of Journal of Functional Biomaterials. This paper can be accepted after moderate revision. The review comments are appended below.
1) In the abstract authors mentioned the amorphous layer thickness 21.813 µm to 222.13 µm. It would be nice if authors include how the thickness has been measured and what instrument or software used for calibrating it.
2) Introduction is well elaborative and clearly defines the research gap. But the nature of implant material and the surface texture of the implant are also crucial to determine the bone joining ability. For instance, the cell anchoring ability of smooth implant surface is poor as compared to the rough implant surface. However, rough implant surface promotes biofilm adhesion. On the other considering the implant material the osteointegration of Ti is far superior to SS316L and Cobalt chromium alloys. In this study bionic function of the implant material has been ruled out. Only the laser treatment procedure was emphasized. It is good to include what is grade of Ti alloy used in this study. Whether it is a pure Ti or Ti-6Al-4V or Ti-Nb alloy or any other. To have some idea on the effect of implant material on bone joining ability please have a look at the following articles.
https://doi.org/10.1016/j.matpr.2022.05.469
https://doi.org/10.1016/j.jallcom.2023.169852
Please do include the implant manufacturers details for better understanding to the readers.
3) How the soft tissue fragments and amorphous layers were distinguished in the histological studies. Since this entire study is only based on histological studies is there any colour difference which could justify the presence of absence of amorphous layer? It would be nice if you could have used different stains for showing amorphous layer and for distinguishing soft tissues. SEM with EDS colour mapping will clearly show the presence or absence of carbonaceous substance in the soft tissues or not. If possible please include it to corroborate your histopathology results.
4) Generally, when a living tissue is exposed to laser then inflammation takes place instantly due to the activation of proinflammatory cytokines. But in the present study the experiments were carried out from a slaughtered animal therefore there may be a chance that thermal side effects such as inflammation is not prominent in the histological analysis which should be clearly justified.
5) In experimental section please do include the laser spot size (i.e., the laser diameter) which is one of the significant parameters determining the area of exposure and the laser interaction time with the soft tissues.
6) There are no details on how the statistical analysis was performed. How values have been calibrated. The results of statistical analysis were not clearly explained in the results and discussion sections.
7) There are lot many typo errors for example: CO2 which is supposed to be CO2 and 2kg/cm2 is supposed to be 2 kg/cm2.
8) The magnification of histological studies is not uniform. Some where the scales were 50µm and some where in group D it was 200µm. It would be nice if the same scales were used for better validation. It is good to mark the areas of smear and fragmented regions to promote better understanding to the readers.
9) Conclusion is too short. It would be nice if the all the results and discussions could be concluded with some scientific understanding.
Author Response
Thanks for the constructive recommendations!
Point 1: In the abstract authors mentioned the amorphous layer thickness 21.813 µm to 222.13 µm. It would be nice if authors include how the thickness has been measured and what instrument or software used for calibrating it.
Response 1: The examination and photomicrodocumentation was performed by using a LEICA DM1000 LED microscope (Germany) and LAS V 4.8 (Leica Application Suite V4, Leica Microsystems, Germany) software. А metric measurement of the smear layer is made, which is based on the difference in hematoxylin and eosin staining. Healthy bone and the smear layer area have different staining intensities and clear boundaries.
Point 2: Introduction is well elaborative and clearly defines the research gap. But the nature of implant material and the surface texture of the implant are also crucial to determine the bone joining ability. For instance, the cell anchoring ability of smooth implant surface is poor as compared to the rough implant surface. However, rough implant surface promotes biofilm adhesion. On the other considering the implant material the osteointegration of Ti is far superior to SS316L and Cobalt chromium alloys. In this study bionic function of the implant material has been ruled out. Only the laser treatment procedure was emphasized. It is good to include what is grade of Ti alloy used in this study. Whether it is a pure Ti or Ti-6Al-4V or Ti-Nb alloy or any other. To have some idea on the effect of implant material on bone joining ability please have a look at the following articles.
Response 2: Thanks for the useful articles, the necessary information has been added.
We added the manufacturers, type of implants and described the conventional technique in material and methodology.
Point 3: How the soft tissue fragments and amorphous layers were distinguished in the histological studies. Since this entire study is only based on histological studies is there any colour difference which could justify the presence of absence of amorphous layer? It would be nice if you could have used different stains for showing amorphous layer and for distinguishing soft tissues. SEM with EDS colour mapping will clearly show the presence or absence of carbonaceous substance in the soft tissues or not. If possible please include it to corroborate your histopathology results.
Response 3: The soft tissues and periosteum from the lower edge of the jaws (corresponding as a feature of the alveolar crest of an edentulous lower jaw) are previously removed. So, in our study, we do not need to distinguished soft tissue fragments and amorphous layers. SEM examination was done. The first results confirm the histological ones, but have not yet been processed and will be the subject of a new article.
Point 4: Generally, when a living tissue is exposed to laser then inflammation takes place instantly due to the activation of proinflammatory cytokines. But in the present study the experiments were carried out from a slaughtered animal therefore there may be a chance that thermal side effects such as inflammation is not prominent in the histological analysis which should be clearly justified.
Response 4: The bone biospecimens were obtained from a regulated slaughterhouse immediately after euthanizing the animals. Treatment was conducted immediately, while the bone still had the characteristics of viable bone. The inflammatory reaction is not the subject of the present study.
Point 5: In experimental section please do include the laser spot size (i.e., the laser diameter) which is one of the significant parameters determining the area of exposure and the laser interaction time with the soft tissues.
Response 5: The program Granulation Tissue Ablation Non-contact was used for the treatment, with the following parameters:
- Laser Energy 400 mJ;
- Pulse Frequency 17 Hz;
- Water Spray Level 6;
- Power 6.80 W.
An AS 7631 (X) Side Firing Tip 1.3 (the laser diameter) x19 mm was used. The interaction time depends on the pulse frequency, which in our study is 17 Hz.
Point 6: There are no details on how the statistical analysis was performed. How values have been calibrated. The results of statistical analysis were not clearly explained in the results and discussion sections.
Response 6: Statistical analysis is the last subtitle in Methods section (right before Results section). There is one paragraph with detailed description of all statistical methods applied. Statistical analysis was performed by a qualified specialist. Standard descriptive statistics was used by presenting the quantitative variables by mean and standard deviation (mean±SD) and Shapiro-Wilk test was applied to inform about the distribution of the units of observation included in the sample. Comparisons between two groups were analyzed with Student’s t-tests for independent samples and between more of two groups with one-way ANOVA with Bonferroni correction for multiple comparisons. A 2-sided p-value of <0.05 was considered statistically significant. Statistical analyses were performed using SPSS Statistics v. 26 software (IBM Corp. Released 2019. Armonk, NY: USA).
In our study, the values of the smear layer after Er:Yag laser treatment (group B) corresponded to the values of the study of Sasaki et al. after laser osteotomy.
Point 7: There are lot many typo errors for example: CO2 which is supposed to be CO2 and 2kg/cm2 is supposed to be 2 kg/cm2.
Response 7: Corrected
Point 8: The magnification of histological studies is not uniform. Some where the scales were 50µm and some where in group D it was 200µm. It would be nice if the same scales were used for better validation. It is good to mark the areas of smear and fragmented regions to promote better understanding to the readers.
Response 8: Measurements in all groups were taken at 50 µm magnification. Figure 5 for group D has been changed we have complied with your recommendation to mark the areas of smear layer.
Point 9: Conclusion is too short. It would be nice if the all the results and discussions could be concluded with some scientific understanding.
Response 9: The smear layer acts as a barrier to the interaction of blood components in the under-lying tissue with the implant surface, leading to delays and complications in the pro-cess of osseointegration. This layer mainly consists of unmineralized collagen and pro-teoglycans. The increase in temperature of the bone surface when working with a ro-tary technique leads to surface carbonization and possible treatment compromising. High hopes are placed on laser ablation and decontamination without additional heat generation due to the precise beam geometry when using short-pulse modes of opera-tion with copious irrigation for cooling. The reduction of the smear layer leads to a tighter contact between the implant surface and the bone, resulting in better primary stability. The cleaning time is reduced, and the osseointegration process is accelerated. The Er:YAG laser program used in our study does not affect the hard tissues and thus does not change the dimensions of the cavity prepared using standard drills.
Obtaining a smeared layer with the conventional technique and calibration inac-curacies in laser ostetomy are minimized with our proposed methodology. The study demonstrated that the combination of rotary technique and subsequent Er:YAG laser treatment of the bone using an original method is a promising prospect in implantolo-gy for achieving faster and stable osseointegration of implants, leading to early func-tional loading.
The synergistic effect of the combination of the two methods leads to an absolutely precisely prepared implant site, which is achieved with conventional rotary methods, and reduced or absent smear amorphous layer on the surface as with the use of laser osteotomy only.
Reviewer 2 Report
I have reviewed the article "Effect of Er:YAG Laser Exposure on the Amorphous Smear Layer in the Marginal Zone of the Osteotomy Site for Placement of Dental Screw Implants: A Histomorphological Study" and found it can be accepted after Major Revision.
What is the conventional method used for the placement of dental screw implants and what is the objective of the study?
What is the Er:YAG laser and how is it used in the study?
How many jaws of euthanized domestic pigs were used in the study and how were the implant osteotomies divided?
What histomorphological techniques were used to measure the reduction in amorphous layer thickness after Er:YAG laser treatment and implant placement?
What is the range of amorphous layer thickness in Group A, which had osteotomy without Er:YAG laser treatment?
What is the range of amorphous layer thickness in Group B, which had osteotomy with Er:YAG laser treatment?
What is the range of amorphous layer thickness in Group C, which had an implant placed in the bone without laser treatment?
Is there a statistically significant difference between the amorphous layer thickness of Group A and Group B, and between the amorphous layer thickness of Group C and Group D?
What are the synergistic effects and the possibility of integrating lasers into the conventional implantation protocol demonstrated in the study?
What is the significance of reducing the smear layer formed during rotary osteotomy using Er:YAG lasers in terms of periimplant space and bone-to-implant contact?
What is the potential impact of accelerated osseointegration facilitated by Er:YAG laser treatment on dental implant success rates?
Please use colorful figures for your study,
To improve the quality of the article, I recommend that the authors review it again and correct the grammatical and spelling errors. They should also improve the article format and consider an appropriate format for the tables and figures. Finally, I suggest that the authors consult some of the following references as a source for comparison to improve the quality of the article:
https://doi.org/10.1016/j.jmrt.2021.03.104
10.22034/NMRJ.2022.04.005
What is the range of amorphous layer thickness in Group D, which had an implant placed after bone treatment with Er:YAG laser?
I have reviewed the article "Effect of Er:YAG Laser Exposure on the Amorphous Smear Layer in the Marginal Zone of the Osteotomy Site for Placement of Dental Screw Implants: A Histomorphological Study" and found it can be accepted after Major Revision.
What is the conventional method used for the placement of dental screw implants and what is the objective of the study?
What is the Er:YAG laser and how is it used in the study?
How many jaws of euthanized domestic pigs were used in the study and how were the implant osteotomies divided?
What histomorphological techniques were used to measure the reduction in amorphous layer thickness after Er:YAG laser treatment and implant placement?
What is the range of amorphous layer thickness in Group A, which had osteotomy without Er:YAG laser treatment?
What is the range of amorphous layer thickness in Group B, which had osteotomy with Er:YAG laser treatment?
What is the range of amorphous layer thickness in Group C, which had an implant placed in the bone without laser treatment?
Is there a statistically significant difference between the amorphous layer thickness of Group A and Group B, and between the amorphous layer thickness of Group C and Group D?
What are the synergistic effects and the possibility of integrating lasers into the conventional implantation protocol demonstrated in the study?
What is the significance of reducing the smear layer formed during rotary osteotomy using Er:YAG lasers in terms of periimplant space and bone-to-implant contact?
What is the potential impact of accelerated osseointegration facilitated by Er:YAG laser treatment on dental implant success rates?
Please use colorful figures for your study,
To improve the quality of the article, I recommend that the authors review it again and correct the grammatical and spelling errors. They should also improve the article format and consider an appropriate format for the tables and figures. Finally, I suggest that the authors consult some of the following references as a source for comparison to improve the quality of the article:
https://doi.org/10.1016/j.jmrt.2021.03.104
10.22034/NMRJ.2022.04.005
What is the range of amorphous layer thickness in Group D, which had an implant placed after bone treatment with Er:YAG laser?
Author Response
Thanks for the constructive recommendations!
Point 1: What is the conventional method used for the placement of dental screw implants and what is the objective of the study?
Response 1: We added the manufacturers, type of implants and described the conventional technique in material and methodology.
Alpha-Bio Neo system (Petach Tikva, Israel) and 3.25 mm x 8.5mm implants of the BT Konic (BTK) system (Italy). They are designed to be active yet gentle on the bone, with an active apical tip and traction wings, providing optimal primary stability while maximizing bone volume preservation. They are made of pure grade four titanium. It provides high technologi-cal characteristics of strength and durability. The surface is heat-etched to create op-timal porosity.
This provides the following benefits:
- Improved early bone-implant contact which is an important factor for excellent primary stability.
- Long-term bone-implant contact.
- Accelerated and improved osseointegration.
- Increased secondary stability.
The implants used are recommended for all types of bone.
Standard preparation of the osteotomy using the implantology surgical set of the respective system was made. The diameter of the final osteotomy drill was 0.1-1.2 mm smaller than the diameter of the implant used. Trepanations were performed using a Bien Air Chiropro implantology unit, a reduction implantology handpiece with con-tinuous external cooling with 0.9% sterile NaCl solution.
Sequence of bone cavity preparation:
- Marking the location to place the implant on the cortical bone using a round bone cutter.
- Trepanation of the bone with a pilot cutter to predetermine the length of the im-plant. Depth gauge check.
- Preparing the implant site with cutters with successively increasing diameters to a diameter by 0.1-1.2 mm smaller than that of the implant. The speed of rotation of the tools is 600-800 revolutions per minute.
- Taking in cortical bone phase with the corresponding profile cutter.
After the cavity is finally prepared, the surface treatment of the walls with the surface treatment module begins, starting with the use of an Er:YAG laser with a pre-ferred wavelength of 2940 nm. The Er:YAG laser (also called erbium-doped yttrium aluminum garnet laser or erbium YAG laser) is a solid-state laser with erbium-doped yttrium aluminum garnet (Er:Y3Al5O12). Er:YAG lasers typically emit infrared light. For the purposes of this study, treatment was performed using a “non-contact granu-lation tissue ablation” program.
The objective of this study was to compare the smear layer thickness of an osteotomy produced using a conventional rotary technique and an osteotomy additionally treated with an Er:YAG laser, according to the method developed by us.
Point 2: What is the Er:YAG laser and how is it used in the study?
Response 2: After the cavity is finally prepared, the surface treatment of the walls with the surface treatment module begins, starting with the use of an Er:YAG laser with a pre-ferred wavelength of 2940 nm. The Er:YAG laser (also called erbium-doped yttrium aluminum garnet laser or erbium YAG laser) is a solid-state laser with erbium-doped yttrium aluminum garnet (Er:Y3Al5O12). Er:YAG lasers typically emit infrared light. For the purposes of this study, treatment was performed using a “non-contact granu-lation tissue ablation” program.
Point 3: How many jaws of euthanized domestic pigs were used in the study and how were the implant osteotomies divided?
Response 3: Mandibles from 10 domestic pigs (Sus scrofa domestica) were examined. Written in material and methodology.
Point 4: What histomorphological techniques were used to measure the reduction in amorphous layer thickness after Er:YAG laser treatment and implant placement?
Response 4: The examination and photomicrodocumentation was performed by using a LEICA DM1000 LED microscope (Germany) and LAS V 4.8 (Leica Application Suite V4, Leica Microsystems, Germany) software. А metric measurement of the smear layer is made, which is based on the difference in hematoxylin and eosin staining. Healthy bone and the smear layer area have different staining intensities and clear boundaries.
Point 5: What is the range of amorphous layer thickness in Group A, which had osteotomy without Er:YAG laser treatment?
Response 5: Histomorphological measurements showed amorphous layer thickness in group A – 21.813 µm to 222.13 µm.
Point 6: What is the range of amorphous layer thickness in Group B, which had osteotomy with Er:YAG laser treatment?
Response 6: Histomorphological measurements showed amorphous layer thickness in group B – 6.08 µm to 43.64 µm.
Point 7: What is the range of amorphous layer thickness in Group C, which had an implant placed in the bone without laser treatment?
Response 7: Histomorphological measurements showed amorphous layer thickness in group C – 5.90 µm to 54.52 µm.
Point 8: Is there a statistically significant difference between the amorphous layer thickness of Group A and Group B, and between the amorphous layer thickness of Group C and Group D?
Response 8: The group comparisons were illustrated in Table 2 and Table 3. There is one sentence (right below each table) that summarized the results presented in the tables and we pointed the appropriate table number after each one.
Point 9: What are the synergistic effects and the possibility of integrating lasers into the conventional implantation protocol demonstrated in the study?
Response 9: The conclusion has been changed at the request of the reviewers and we have explained to it what the synergistic effect is.
The synergistic effect of the combination of the two methods leads to an absolutely precisely prepared implant site, which is achieved with conventional rotary methods, and reduced or absent smear amorphous layer on the surface as with the use of laser osteotomy only.
Point 10: What is the significance of reducing the smear layer formed during rotary osteotomy using Er:YAG lasers in terms of periimplant space and bone-to-implant contact?
Response 10: In conclusion . The smear layer acts as a barrier to the interaction of blood components in the under-lying tissue with the implant surface, leading to delays and complications in the pro-cess of osseointegration. This layer mainly consists of unmineralized collagen and pro-teoglycans. The increase in temperature of the bone surface when working with a ro-tary technique leads to surface carbonization and possible treatment compromising. High hopes are placed on laser ablation and decontamination without additional heat generation due to the precise beam geometry when using short-pulse modes of opera-tion with copious irrigation for cooling. The reduction of the smear layer leads to a tighter contact between the implant surface and the bone, resulting in better primary stability. The cleaning time is reduced, and the osseointegration process is accelerated. The Er:YAG laser program used in our study does not affect the hard tissues and thus does not change the dimensions of the cavity prepared using standard drills.
Obtaining a smeared layer with the conventional technique and calibration inac-curacies in laser ostetomy are minimized with our proposed methodology. The study demonstrated that the combination of rotary technique and subsequent Er:YAG laser treatment of the bone using an original method is a promising prospect in implantolo-gy for achieving faster and stable osseointegration of implants, leading to early func-tional loading.
Point 11: What is the potential impact of accelerated osseointegration facilitated by Er:YAG laser treatment on dental implant success rates?
Response 11: : In conclusion .The smear layer acts as a barrier to the interaction of blood components in the under-lying tissue with the implant surface, leading to delays and complications in the pro-cess of osseointegration. This layer mainly consists of unmineralized collagen and pro-teoglycans. The increase in temperature of the bone surface when working with a ro-tary technique leads to surface carbonization and possible treatment compromising. High hopes are placed on laser ablation and decontamination without additional heat generation due to the precise beam geometry when using short-pulse modes of opera-tion with copious irrigation for cooling. The reduction of the smear layer leads to a tighter contact between the implant surface and the bone, resulting in better primary stability. The cleaning time is reduced, and the osseointegration process is accelerated. The Er:YAG laser program used in our study does not affect the hard tissues and thus does not change the dimensions of the cavity prepared using standard drills.
Obtaining a smeared layer with the conventional technique and calibration inac-curacies in laser ostetomy are minimized with our proposed methodology. The study demonstrated that the combination of rotary technique and subsequent Er:YAG laser treatment of the bone using an original method is a promising prospect in implantolo-gy for achieving faster and stable osseointegration of implants, leading to early func-tional loading.
Point 12: Please use colorful figures for your study. To improve the quality of the article, I recommend that the authors review it again and correct the grammatical and spelling errors. They should also improve the article format and consider an appropriate format for the tables and figures.
Response 12: We have followed the example articles and added more figures.
Point 13: What is the range of amorphous layer thickness in Group D, which had an implant placed after bone treatment with Er:YAG laser?
Response 13: Histomorphological measurements showed amorphous layer thickness in group D – 1.29 µm to 6.98 µm.
Round 2
Reviewer 1 Report
Authors have satisfactorily amended all the review suggestions. Therefore, the manuscript can be accepted.
Author Response
We are grateful for your comments!
We agree with them and made the appropriate corrections.
- We correct the reference style within the text,
- We correct the omission of the definite article within the text and minor typos.
Reviewer 2 Report
Paper can be accepted in this form.
Paper can be accepted in this form.
Author Response

(The authors gave the same response as above.)
